# Monte Carlo Tree Search With Iteratively Refining State Abstractions

**Samuel Sokota**
Carnegie Mellon University
ssokota@andrew.cmu.edu

**Caleb Ho**
Independent Researcher
caleb.yh.ho@gmail.com

**Zaheen Ahmad**
University of Alberta
zfahmad@ualberta.ca

**J. Zico Kolter**
Carnegie Mellon University
zkolter@cs.cmu.edu

## Abstract

Decision-time planning is the process of constructing a transient, local policy with the intent of using it to make the immediate decision. Monte Carlo tree search (MCTS), which has been leveraged to great success in Go, chess, shogi, Hex, Atari, and other settings, is perhaps the most celebrated decision-time planning algorithm. Unfortunately, in its original form, MCTS can degenerate to one-step search in domains with stochasticity. Progressive widening is one way to ameliorate this issue, but we argue that it possesses undesirable properties for some settings. In this work, we present a method, called abstraction refining, for extending MCTS to stochastic environments which, unlike progressive widening, leverages the geometry of the state space. We argue that leveraging the geometry of the space can offer advantages. To support this claim, we present a series of experimental examples in which abstraction refining outperforms progressive widening, given equal simulation budgets.

## 1 Introduction

In control problems, an agent makes sequential decisions toward the end of accumulating a large amount of reward [21]. Many reinforcement learning algorithms approach this task by computing a policy, which, given a state, dictates the agent's behavior; others operate under the paradigm of decision-time planning, in which, after the agent has arrived at a state, it spends additional computation constructing or revising its policy for the immediate decision(s). The decision-time planning paradigm can be advantageous because it allows for an additional step of policy improvement that is not available to agents acting according to a predetermined policy.

Of the numerous ways to perform decision-time planning, Monte Carlo tree search (MCTS) is among the most influential [6, 10, 15]. Algorithms powered by MCTS, like AlphaZero and MuZero, yield strong performance in games with complex value function landscapes, like Go, chess, Hex, and shogi [2, 20, 19]. MuZero even shows strong performance on Atari games, where model-free algorithms had previously been the most performant.

Unfortunately, in its original form, MCTS can degenerate to a one-step search when the transition function has an infinite support. Even if the support is finite but non-trivial, MCTS may be relegated to building shallow search trees. However, despite the importance of stochasticity in real-world problems, relatively little attention has been paid to this issue, perhaps as a result of the fact that many popular benchmarks are deterministic or close-to-deterministic.

35th Conference on Neural Information Processing Systems (NeurIPS 2021).

One way to address this issue is by using progressive widening [9]. Progressive widening works by alternating between adding new children and selecting among existing children such that, asympototically, the search tree becomes both infinitely deep and infinitely wide. Progressive widening is also advantageous because it includes hyperparameters that control its propensity to add new children (as opposed to selecting among existing children). These hyperparameters can interpolate between width-wise expansion only (vanilla MCTS) and depth-wise expansion only (transition determinization). As a result, with proper tuning, progressive widening is capable of performing well in both settings with important stochasticity and in settings with unimportant (or non-existent) stochasticity.

However, progressive widening's decision rule for expansion is the same for every transition. In settings with non-uniform stochasticity, we argue that this property is undesirable, as it requires progressive widening to compromise between focusing on unimportant stochasticity and ignoring important stochasticity.

In this work, we propose a new method, called abstraction refining, for extending MCTS to stochastic settings. Abstraction refining uses random and iteratively refining state abstractions defined by the geometry of the state space. If a newly sampled state is similar to a state that is already in the tree, it is discarded in favor of the pre-established state; otherwise, the newly sampled state is added to the tree. Because the criteria for similarity becomes more strict the more often an abstraction is used, abstraction refining guarantees that, in the limit, the search tree will grow both infinitely wide and infinitely deep.

In addition to proposing the abstraction refining algorithm, our contributions are twofold. First, we present both a proof that, when used for policy evaluation, abstraction refining converges almost surely in finite MDPs. Second, we make an empirical case that, provided a good notion of state similarity, abstraction refining can outperform progressive widening in settings with stochasticity of varying importance, given an equal simulation budget. To make this case, we present a series of domains in which i) stochasticity is sometimes, but not always, relevant to performing well on the task and ii) the naive notion of distance between states is reflective of behavioral similarity (though we also include some settings in which we use a learned notion of distance). In these domains, we find that abstraction refining can outperform progressive widening.

## 2 Background

We consider a Markov decision process (MDP) setting $\langle \mathcal{S}, \mathcal{A}, \mathcal{T}, \mathcal{R}, \gamma \rangle$, with state space $\mathcal{S}$, action space $\mathcal{A}$, transition function $\mathcal{T}$, reward function $\mathcal{R}$ and discount factor $\gamma$. The objective is to determine a policy $\pi$ that achieves a large expected cumulative reward $\mathbb{E}_\pi \left[ \sum_{t=0}^\infty \gamma^t \mathcal{R}(s_t, a_t) \right]$. Rather than explicitly computing a function $\pi$, decision-time planning algorithms' policies are defined implicitly by their planning procedures. These planning procedures are given generative access to (i.e., the ability to sample from) the transition function $\mathcal{T}$.

We also consider a Markov reward process (MRP) setting $\langle \mathcal{S}, \mathcal{T}, \mathcal{R}, \gamma \rangle$, with state space $\mathcal{S}$, transition function $\mathcal{T}$, reward function $\mathcal{R}$, and discount factor $\gamma$. An MRP is like an MDP, except that the agent is a passive observer, rather than a participator. Equivalently, one could also think of an MRP as an MDP in which the policy has been fixed and baked into the transition and reward functions. For the purposes of this work, the goal of an agent in an MRP is to evaluate the expected return from state—i.e., to perform policy evaluation.

### 2.1 Monte Carlo Tree Search

MCTS works by iteratively building a look-ahead tree of states. For each action $a$ at a state $s$, the algorithm keeps track of the number of times the action has been selected at that state $N(s, a)$ and the average of the value assessments of that action $Q(s, a)$. At each iteration, the agent (i) *selects* a leaf node to expand; (ii) *expands* the tree by adding a child node of the leaf node; (iii) *evaluates* the newly added state, and (iv) *backpropagates* the evaluation from the new child to the root.

**Selection** In the selection phase, starting from the root node, which corresponds to the state $s_0$ currently occupied by the agent, MCTS traverses the tree until a leaf node is reached. The tree traversal is dictated by a *selection policy*. The most commonly used selection policies are based on the UCT algorithm [15], or its variants, like PUCT [18]. Although it is less common, MCTS can also be used as a policy evaluation algorithm. In this case, the selection policy is set to the target policy.

**Expansion** Once a leaf node is reached, the tree is expanded by adding a child of the leaf node to the tree. If the leaf node is a terminal state, no children are added.

**Evaluation** Whenever MCTS adds a new state to the tree, it evaluates the state. Traditionally, MCTS practitioners performed evaluation by generating a rollout to the end of the game using a fast-to-evaluate simulation policy. Modern implementations often use a parameterized value function in lieu of simulation rollouts.

**Backpropagation** After MCTS evaluates a state, it updates the nodes along the path from the root to the evaluated state. Denote the path from the root to the leaf using $s_0, a_0, r_0, \ldots, s_T$. Then for each pair $(s_k, a_k)$, MCTS updates $Q(s_k, a_k)$ using a weighted average between the existing value $Q(s_k, a_k)$ and the leaf evaluation summed with the accumulated reward $r_{k+1} + \cdots + r_T$. Additionally each visit counter $N(s_k, a_k)$ gets incremented by one.

**Selecting an Action To Execute** After the search is over, MCTS selects an action for the agent to take. Common ways of doing this include selecting the action at the root node with the highest average $Q$-value, the action with highest lower confidence value or the action with the highest visit count. When used in an evaluation context, MCTS returns the average value of the root node.

**What about Stochasticity?** The description of MCTS above assumes that the environment is deterministic. In games with stochasticity, the selection process has an additional complication. Specifically, after an action $a$ is selected at a state $s$, there is no uniquely determinable next state $s'$, but rather, a distribution over next states $\mathcal{T}(s, a)$. The most obvious way to extend MCTS to this setting is to sample $s' \sim \mathcal{T}(s, a)$. If there is already a child node for $s'$, then that node is selected and the selection process continues. Otherwise, if there is no child node for $s'$, one is added to the tree and evaluated. The consequence of this modification is that, for a fixed number of iterations, the search tree cannot be as deep as it would be in a deterministic setting. In the most extreme case, where there are an infinite number of possible next states, this modification can degenerate MCTS into one-step search (because it may be the case that no state is sampled twice). This means that the performance of MCTS in games in which deep search trees are crucial for good performance could be ruined by arbitrarily small (but infinite support) perturbations to state representations.

## 2.2 Progressive Widening

Progressive widening [9] is one way to augment MCTS in such a way that it can build deep search trees in arbitrarily stochastic domains.[1] The basic idea behind progressive widening is to alternate between sampling new next states and sampling among the next states that are already in the tree. If the state-action pair $N(s, a)$ has been tried a large number of times relative to the number of successor states in the tree, progressive widening adds a new state. On the other hand, if there are a large number of successor states relative to the number of times $N(s, a)$ has been tried, progressive widening samples among the successor states already in the tree. Whether or not to sample is determined by the boolean num_children$[s, a] < k \cdot N(s, a)^\alpha$ where $\alpha \in [0, 1]$ and $k \in \mathbb{R}_+$ are hyperparameters. Simplified pseudocode for the sampling step of progressive widening is shown in Algorithm 1.

Informally, the hyperparameter $\alpha$ can be thought of as a propensity to select among existing children, rather than adding new children. When $k = 1$, $\alpha = 0$, progressive widening reduces to transition determinization—the first time that a state-action pair is visited, a successor state is sampled and added to the tree; thereafter, the same successor state is selected every time the state-action pair is visited. When $k = 1$, $\alpha = 1$, progressive widening reduces to vanilla MCTS—every time a state action pair is encountered, a new successor state is sampled and added to the tree.

If $k$ and $\alpha$ can be properly tuned, progressive widening offers flexibility. In domains in which stochasticity is important, $\alpha$ can be set to one or close to one. In domains in which stochasticity can be ignored, $\alpha$ can be set to zero or close to zero. Otherwise, an intermediate value often works well.

However, there is a downside to progressive widening. In particular, it cannot discriminate between stochasticity that matters and stochasticity can safely be ignored. This means that, if some stochasticity

---

[1]Note that we call progressive widening here is generally referred to as double progressive widening. Double progressive widening uses progressive widening over both the action space and the state space. For simplicity, this work restricts its attention to small action spaces, and therefore uses the term progressive widening to refer to the state space behavior.

is important but some is unimportant, the best thing that progressive widening can do requires either placing too much emphasis on unimportant stochasticity, or too little on important stochasticity.

---

**Algorithm 1** Progressive Widening

> **procedure** SAMPLE($s, a$)
>   $N \leftarrow$ num_visits$[s, a]$
>   **if** num_children$[s, a] < kN^\alpha$ **then**
>     $s' \sim \mathcal{T}(s, a)$
>     **if** $s' \notin$ children$[s, a]$ **then**
>       children$[s, a]$.add($s'$)
>     return $s'$
>   **else**
>     return sample(children$[s, a]$)

---

**Algorithm 2** Abstraction Refining

> **procedure** SAMPLE($s, a$)
>   $s' \sim \mathcal{T}(s, a)$
>   $s'' \leftarrow$ nearest_neighbor($s'$, children$[s, a]$)
>   **if** $d(s', s'') < \epsilon_{\text{num\_visits}[s,a,s'']}$ **then**
>     return $s''$
>   **else**
>     children$[s, a]$.add($s'$)
>     return $s'$

---

# 3 Abstraction Refining

In this work, we introduce an alternative to progressive widening that we call abstraction refining. Abstraction refining is motivated by the deficiency of progressive widening described above: in settings with heterogeneous stochasticity, progressive widening is forced to make an undesirable compromise. Rather than defining expansion rate purely based on visit-count, as progressive widening does, abstraction refining determines whether or not to expand using state similarity. At each selection step, abstraction refining samples a new state and checks if it is similar to an existing child. If it is, abstraction refining abstracts them together and selects the similar, existing child; if it is not, abstraction refining adds the new state to the tree. In doing so, in principle, abstraction refining avoids the weakness of progressive widening: when stochasticity is unimportant, abstraction refining can cluster states together but when it is important, abstraction refining can cluster states separately. Simplified pseuodocode for abstraction refining is given in Algorithm 2.

Each state abstraction is defined by two principles. The first principle is that, if a state is to be abstracted, it should be abstracted to its nearest neighbor among states already in the tree. Equivalently, this principle states that, when abstractions take place, they should use a mapping induced by the Voronoi diagram over the children in the tree.

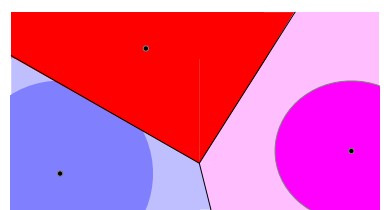

Figure 1: A visual depiction of an example low dimensional state space. There are three states already in the tree, represented by the black dots. The Voronoi diagram is shown using the colors blue, red and purple. The intersection of the Voronoi cells with each child's epsilon ball (shaded darker), corresponds to states that would be abstracted if they were sampled. The lighter shading shows the states that would be added to the tree if they were sampled.

The second idea is that, if abstraction is used many times, its notion of similarity should become more precise. This idea is motivated by controlling the amount of error that is introduced by a state abstraction. A necessary condition for asymptotic correctness is that state abstractions become arbitrarily small in the limit. Toward that end, abstraction refining makes use of a sequence $\{\epsilon_n\}_{n\in\mathbb{N}}$ that is strictly decreasing ($\epsilon_n > \epsilon_m$ for $n < m$) and going to zero in the limit ($\lim_{n\to\infty} \epsilon_n = 0$). A state that has been selected $n$ times previously can only abstract new states within a distance of $\epsilon_n$.

A low dimensional example is shown in Figure 1.

## 3.1 Consistency

Used for policy evaluation in a finite MDP, abstraction refining is asymptotically consistent.

**Proposition 1.** *Let $\langle \mathcal{S}, \mathcal{R}, \mathcal{T} \rangle$ be a finite Markov reward process with horizon $h$ and bounded reward. Then for any $s \in \mathcal{S}$, abstraction refining's assessment of state $s$ converges to $v(s)$, almost surely, where $v(s) = \mathbb{E}\left[\sum_{t=d}^{h-1} \mathcal{R}(S_t) \mid S_d = s\right]$, given a bounded evaluation function.*

*Proof.* We proceed by induction. First, consider that for any state $s_{h-1}$, abstraction refining converges surely after one iteration because $v(s_{h-1}) = \mathcal{R}(s_{h-1})$. Now consider some state $s_d$ at arbitrary depth $d$. The value assessed by abstraction refining after $n$ iterations is $V_n(s_d) = \mathcal{R}(s_d) + \sum_{s \in \mathcal{S}_{d+1}} \frac{N_{s,n}}{n} V_{N_{s,n}}(s)$ where $N_{s,n}$ denotes the number of times that state $s$ has been visited by the $n$th iteration. By Lemma 1, we have that $N_{s,n}/n$ converges to $\mathcal{T}(s \mid s_d)$ almost surely. Note that by Lemma 2, and by inductive hypothesis, $V_{N_{s,n}}(s)$ converges to $v(s)$ almost surely. Then invoking the fact that the term-wise product of sequences of almost surely convergent random variables converges almost surely to the product of the limits, and the fact that the term-wise sum of sequences of almost surely convergent random variables converges almost surely to the sum of the limits, and Lemma 3, we conclude that $V_n(s_d)$ converges to $\mathcal{R}(s_d) + \sum_{s \in \mathcal{S}_{d+1}} \mathcal{T}(s \mid s_d) v(s) = v(s_d)$ almost surely. $\square$

Proofs for the lemmas below can be found in the appendix.

**Lemma 1.** *For a fixed $s$, the sequence $N_{s,n}/n$ converges almost surely to $\mathcal{T}(s \mid s_d)$.*

**Lemma 2.** *For a fixed $s$ with $\mathcal{T}(s \mid s_d) > 0$, the sequence $N_{s,n}$ diverges almost surely.*

**Lemma 3.** *Let $X_n$ be a sequence converging to $x$ almost surely. Let $f \colon \mathbb{N} \to \mathbb{N}$ be a monotonically increasing, onto function. Then $Y_n = X_{f(n)}$ converges to $x$ almost surely.*

Given Proposition 1, the extension to the discounted infinite horizon is relatively straightforward.

**Corollary 1.** *Abstraction refining also converges almost surely in a policy evaluation setting for infinite horizon finite Markov reward processes, with rewards discounted by $\gamma \in [0, 1)$.*

*Proof.* (Sketch) Fix $\epsilon > 0$. Select $h$ such $\sum_{t=h}^{\infty} \gamma^t m < \epsilon/4$, where $m$ is a bound on the reward and evaluation function. Then the amount of error that can be accrued from states beyond time $h$ is bounded by $\epsilon/2$, surely. Additionally, by Proposition 1, the amount of error that can be accrued over a finite horizon is bounded by $\epsilon/2$ for all sufficiently large $n$, almost everywhere. Therefore, the aggregate error is bounded by $\epsilon$, almost everywhere for all sufficiently large $n$. $\square$

It is important to note that convergence in the policy evaluation case does not guarantee convergence in combination with tree search algorithms [16]. In contrast to abstraction refining, there do exist variants of progressive widening for which such guarantees have been proven [5]. In other words, the theoretical justification for abstraction refining presented here is less strong than the existing theoretical justification for progressive widening.

## 3.2 Runtime Discussion

In a decision-time planning paradigm, which is where MCTS is most often used, planning time is scarce. Thus, the runtime of algorithmic extensions to MCTS is important. However, high performance MCTS implementations almost ubiquitously use large amounts of parallelization [8]. As a result, the most important analysis regards non-parallelizable runtime. We discuss abstraction refining's runtime for both serial and parallelized cases.

Abstraction refining performs two potentially expensive computations. The first of these is sampling states $s' \sim \mathcal{T}(s, a)$, which could be costly if the simulator is slow, or if sampling is being executed from a large learned model. Abstraction refining must perform this sampling at each state it visits during its selection process. In contrast, progressive widening only needs to perform sampling once each selection process. Thus, in settings where sampling states is a bottleneck, abstraction refining could add a significant amount of additional overhead on top of progressive widening. However, if sampling can be parallelized, this cost can be amortized by sampling a large number of next states in parallel the first time a state-action pair is visited.

The second potentially expensive computation is computing the nearest neighbor. This operation takes linear time serially (in the number of children) and logarithmic time when fully parallelized, and

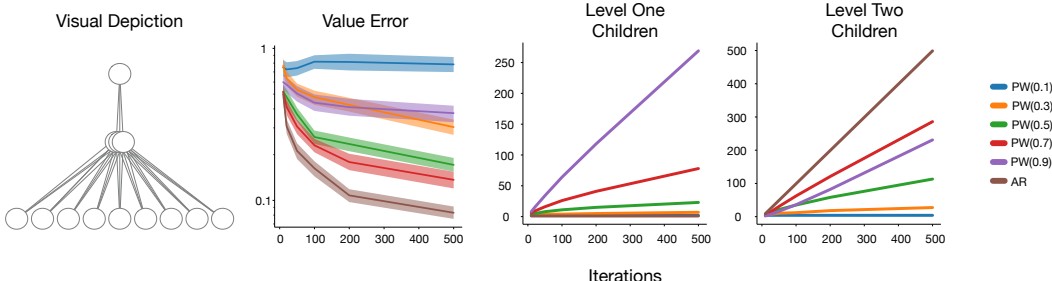

Figure 2: A comparison between progressive widening and abstraction refining on a toy MRP designed to illustrate the drawback of progressive widening's global hyperparameters. (far left) A visualization of the MRP; (middle left) value error, shown on a log scale, as a function of iterations; (middle right) number of nodes at depth one of the tree as a function of the number of iterations; (far right) number of at depth two of the tree as a function of the number of iterations.

must be performed at each step of the selection process. In contrast, progressive widening requires no such operation. Therefore, for systems in which a nearest neighbor computation would be the bottleneck for MCTS, abstraction refining would be substantially more expensive per iteration than progressive widening. On the other hand, for systems in which this would not be the case, perhaps such as those that use large networks to perform evaluations, abstraction refining may offer a cost per iteration competitive with that of progressive widening.

In our experiments, we compare abstraction refining and progressive widening given equal numbers of search iterations. It is worth bearing in mind, given the discussion above, that this is not always a fair comparison.

## 4    Experiments

We present four sets of experiments in the main body of the paper. The first two investigate the prediction (policy evaluation) problem; the second two investigate the control problem. A fifth set of experiments, as well as complementary experiments to those in the main body of the paper, are included in the appendix. In all plots, bands depict estimates of 95% confidence intervals computed using bootstrapping, except those in the pendulum plots in the appendix, which were computed using the central limit theorem.

**Illustrating Progressive Widening's Weakness**    For our first experiment, we seek to confirm the claim made in previous parts of the paper—that progressive widening is forced to make an undesirable compromise in situations with heterogeneous stochasticity. Toward that end, we construct a toy Markov reward process, depicted visually in Figure 2 (far left). In the MRP, the agent begins in the single state at the top. The first transition moves the agent to one of an infinite number of very similar states (level one), shown in the middle. At these states, the reward function is zero and the transition function is nearly identical. The second transition moves the agent to one of an infinite number of dissimilar states (level two), shown at the bottom. The agent is given a bad value function for the level one states and a good value function for the level two states, reflecting a situation in which searching deeply is important. To evaluate the expected return in the fewest number of iterations, the optimal behavior for a state selector is to add as few level one children as possible and as many level two children as possible.

This problem is difficult for progressive widening to address. For $k = 1$, setting $\alpha$ to a small value restricts the number of level one children but also restricts the number of level two children (see blue and orange in middle right and far right plots; Figure 2). But setting $\alpha$ to a large value causes progressive widening to repeatedly add a large number of level one children, preventing focus on level two children (see purple in middle and far right plots; Figure 2). Intermediate values of $\alpha$ (see red and greed in middle and far right plots; Figure 2) offer the best compromise but, nevertheless, add too many level one children and too few level two children.

In contrast, abstraction refining is perfectly suited to address this problem—it can abstract the very similar level one states together while treating the dissimilar level two states separately using

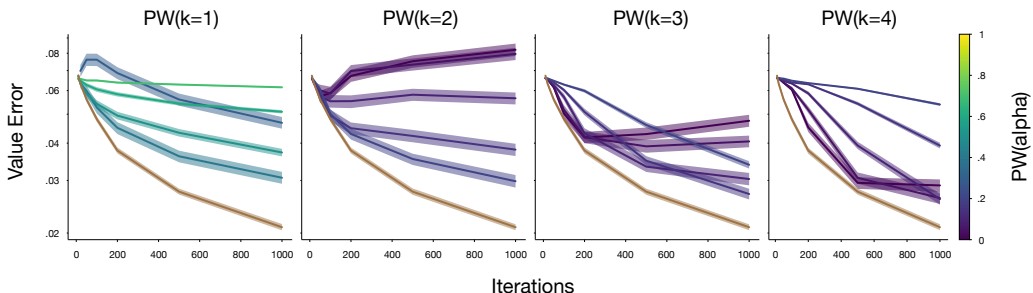

Figure 3: Abstraction refining (brown), with a fixed set of hyperparameters, compared against progressive widening with various hyperparameter values on policy evaluation in a continuous variant of blackjack. The value error (y-axis) is shown on a log scale.

$\epsilon_n = n^{-0.1}$. As a result, it can spend its entire quota on level two states (see brown in middle and far right plots; Figure 2). These distinctions manifest directly in prediction error, as shown in Figure 2, middle left plot, where abstraction refining outperformed progressive widening over each $\alpha$ value.

We also ran these experiments using other values of $k$ for progressive widening and found qualitatively similar results. For these plots and a specification of the MRP, see the appendix.

**Policy Evaluation in Blackjack**    The previous section illustrated that, in a policy evaluation setting designed adversarially for progressive widening, abstraction refining can offer superior per iteration performance. To confirm that the qualitative takeaway of the previous section was not solely a result of the adversarial design, this section investigates policy evaluation on a less contrived setting—a variant of blackjack with continuous card values. We chose a policy described by *Wizard of Odds Consulting*. Details for the game, the policy, and plots with additional hyperparameters can be found in the appendix.

The results are shown in Figure 3. Abstraction refining with $\epsilon_n = 2n^{-0.1}$ is shown in brown in each subplot. The $k$ value of progressive widening increases from one to four from left to right. Darker colors correspond to smaller lighter values of alpha. An effort was made to prune noncompetitive $\alpha$ values from the graph to reduce visual clutter.

There are a number of observations to make. First, notice the relationship between $k$ and $\alpha$ for progressive widening. As $k$ becomes the larger, the most competitive values of $\alpha$ become smaller, reflecting the fact that $k$ also plays a role in progressive widening's propensity to expand its width. In general, the role $k$ plays is subtle—for $k > 1$, the number of children at a particular grows as $\min(n, kn^\alpha)$. This means that, for the first $k^{1/(1-\alpha)}$ times a node is visited, it always adds a child but, thereafter, only adds a new child a much smaller proportion of the time.

Second, notice the instability that can result from small $\alpha$ values. This reflects the reality that small $\alpha$ values can build deep trees quickly, which can be advantageous, but expand slowly width-wise, leading to repeated evaluations of states that do not accurately reflect the distribution.

Lastly, notice that while some settings of progressive widening perform competitively with abstraction refining for some iteration numbers, there is no setting that performs competitively across all iterations. In other words, progressive widening does not perform competitively with abstraction refining on an *anytime* basis. While this is not necessarily problematic in a strictly policy evaluation context, it is an undesirable for a subcomponent of MCTS, used for control, for two reasons: first, because MCTS uses previous evaluations to inform its selection decisions, performing inaccurate evaluations at some iteration horizon could cause it to misallocate its budget to the wrong actions; second, because MCTS visits different actions and states vastly different numbers of times in a fashion that is not initially predictable, there is no good way to determine the iteration horizon for which to optimize.

**Control in Trap**    Thus far, we have focused on comparing state selectors in a policy evaluation setting. While such comparisons make sense, as state selectors serve an evaluatory purpose, MCTS is rarely used for policy evaluation. As a result, the utility of a state selector is closely linked to its ability to increase performance on control tasks. For our first control experiment, we use a variant of the trap

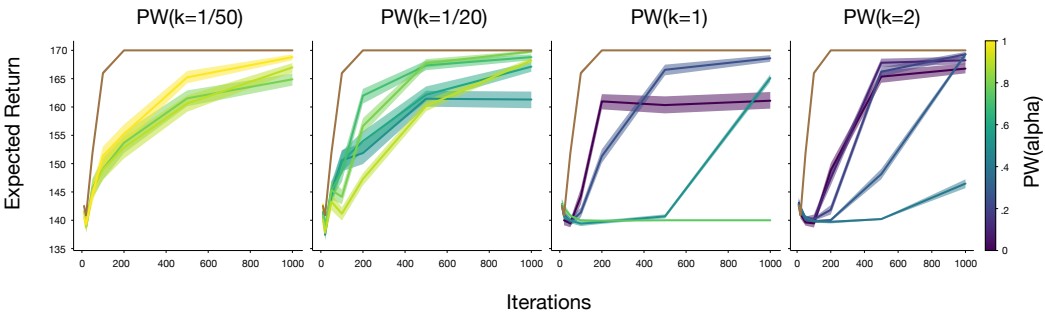

Figure 4: Abstraction refining (brown), with a fixed set of hyperparameters, compared against progressive widening with various hyperparameter values on control in a variant of the trap problem.

problem, which was first introduced in the original progressive widening paper [9] to demonstrate the advantages of progressive widening compared to vanilla MCTS. The basic idea of problem is that an agent must move from one platform to another without falling into the trap. Stochasticity comes in the form of actuator error—the amount the agent intends to move differs from the amount the agent actually moves by stochastic noise. A specification of the problem, hyperparameter settings, and plots for additional hyperparameter values can be found in the appendix.

The results are shown in Figure 4. Abstraction refining with $\epsilon_n = 0.1n^{-0.1}$ is shown in brown in each subplot. Again, the $k$ value of progressive widening is shown increasing from left to right, the $\alpha$ value is depicted by hue, with darker values meaning smaller $\alpha$s, and the plots were pruned of non-competitive results to result visual clutter.

In the results, we see that a single hyperparameter configuration of abstraction refining is able to outperform progressivew widening across a wide range of hyperparameters. One reason for abstraction refinement's strong performance here is likely that it is able to discriminate between having fallen into the trap and not having fallen into the trap. On the other hand, progressive widening has no mechanism for making this distinction, which may lead to it being overly confident about the safety of large jumps. In the appendix, we show additional results for abstraction refining with a distance function that does not allow for such a distinction. We find that, given this distance function, abstraction refining is unable to reproduce the level of performance shown in Figure 4.

**Opponent Exploitation in Five by Five Go**   For our fourth experiment, we use AlphaZero to compare abstraction refining and progressive widening in opponent exploitation in five by five Go. In our setup, the first moving player (Black) is controlled by an AlphaZero policy network. The second moving player (White) performs search using the latent state representation of the AlphaZero network as its state representation of the system. Because Black's policy is fixed, White can view the game as a stochastic MDP in which the transition function is dictated by Black's policy. We also add a small amount of noise to the observation function to increase the difficulty of the search problem. We quantify performance using the amount of territory controlled by a player at the end of a game (i.e., the margin of victory), rather than the win-loss value. We modified OpenSpiel's [17] AlphaZero and Go implementations for this experiment. Discussion of these design choices and the experimental setup can be found in the appendix.

Results are shown in Figure 5, where $k = 1$ for progressive widening. While the results are somewhat uncertain, there appears to be upward trends in $\alpha$, suggests that larger values of $\alpha$ perform better, and in the number rollouts, suggesting that larger rollout budgets perform better, as expected. It seems to be the case that abstraction refining is able to outperform progressive widening—possibly because it is able to abstract together board states that are behaviorally identical but differing superficially by the stochastic noise. In contrast, progressive widening is forced to treat these states separately.

**Control in Pendulum**   An additional set of experiments performed on a variant of pendulum in which the agent experiences actuator error are included in the appendix.

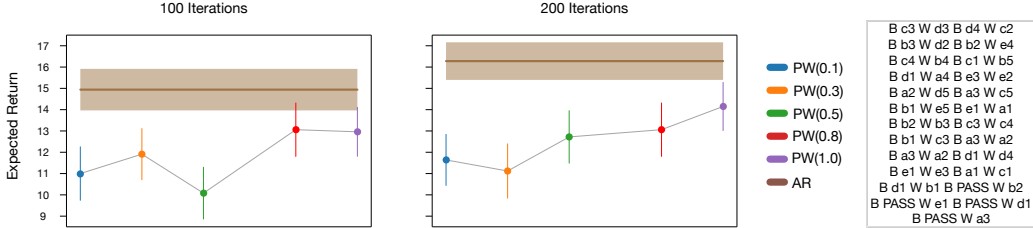

Figure 5: (left) Abstraction refining (brown), compared against progressive widening with $k = 1$ and various $\alpha$ values for opponent exploitation in five by five Go; (right) an example game in which the exploiter wins by the maximum margin (25).

## 5 Closing Discussion

**On The Distance Function** An important facet of abstraction refining that, heretofore, has gone largely uncommented upon, is that it requires a notion of distance between states in order to compute the nearest neighbor. It is readily apparent that using Euclidean distance over the canonical state representations, as we did in our first three experiments, would perform poorly in many cases. And while our 5x5 Go and pendulum experiments (the latter of which are included in the appendix) offer some initial work suggesting that using abstraction refining over learned state embedding can be successful, it would be better to have a more principled way for constructing this distance metric. Perhaps the most natural notion of state similarity for abstraction refining is a bisimulation metric [12, 11]. Informally, under a bisimulation metric, two states are similar if, for all actions, (i) the states yield similar rewards and (ii) the states have similar transition distributions (defined recursively in terms of the bisimulation metric)—in short, if they are behaviorally similar. However, while there has been recent progress in learning bisimulation metrics [24, 7], doing so in large MDPs with arbitrary transition distributions remains a challenging problem.

**Related Work** There are a number of works related to this one in that they use a notion of distance between actions to limit the branching factor and improve action selection. Ahmad et al. show how to leverage Delaunay triangulation for action selection in a single-decision setting [1]. Yee et al. propose incorporating kernel regression into MCTS for continuous action spaces [23]. Most recently, Kim et al. show how Voronoi diagrams can be applied to extend MCTS to a continuous action space [14]. Our work can be viewed as following in the spirit of that of Kim et al., but focused on states (and evaluations) rather than actions (and control).

There has also been work leveraging state similarity for MCTS. Specifically, Xiao et al. propose using kernel regression to improve state evaluations for MCTS [22]. Experimentally, they show that using an intermediate layer of AlphaZero's policy-value network in 19x19 Go to compute similarity performs well in practice.

Outside of progressive widening [9], there also additional work on performing state abstractions during search [13]. Specifically, Hostetler et al. investigate two mechanisms for adapatively refining state abstractions. The first, called random refinement, randomly divides existing state abstractions. The second, called decision tree-based refinement, uses a decision tree over feature space of the successor states to determine state abstractions, and adds new splits to the tree for refinement.

Finally, there is also additional work motivated by the deficiencies of MCTS in settings with large branching factors arising from stochasticity [3, 4]. Anthony proposes performing additional policy gradient updates to the network to improve the policy during decision time, rather than maintaining a tabular tree of visited nodes. Results suggest that this approach, called policy gradient search, can achieve performance comparable or superior to that of MCTS in Hex.

**Summarizing Remarks** This work introduces abstraction refining, a new method for performing state selection for MCTS in stochastic environments. We prove that abstraction refining converges almost surely when used for policy evaluation in finite MDPs and, through a series of experiments, show evidence suggesting that abstraction refining can offer advantages over progressive widening in settings with stochasticity of varying importance, given an equal number of simulations.

## Acknowledgments and Disclosure of Funding

We thank Ryan D'Orazio, Alexander Robey and Yiding Jiang for providing helpful feedback for this work.

This work was supported by funding from the Bosch Center for Artificial Intelligence.

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
