# OpenReview forum: "Monte Carlo Tree Search With Iteratively Refining State Abstractions"
_NeurIPS.cc/2021/Conference — NeurIPS 2021 Poster_

### Official Review · Reviewer_b8VZ · 2021-07-10

**Rating:** 6
**Confidence:** 4

**Summary:**

This paper proposes a method, called abstraction refining, for extending MCTS to stochastic environments that leverages the geometry of the state space. Compared to progressive widening with global hyperparameters for every state, abstraction refining groups the states together dynamically depending on each state. Their results show abstraction refining outperforms progressive widening in the way of their experiments of this paper.

**Limitations And Societal Impact:**

Yes.

**Main Review:**

While it is interesting for the proposed method to reduce the large search space like progressive widening, there are the following major concerns.
This paper gives four examples in the experiments, but experimental analyses are most based on value errors. It is not convincing in this way since the most important criteria for MCTS is the strength, not just value errors. This paper still needs more experiments including the strength to demonstrate the idea is working.
In addition, in these examples, the nearest neighbors are calculated all based on the geometry of state  space (or Euclidean distance). It is very hard to convince people to believe that this would work for some games, like Go. This paper mentions we could use better distance functions for the nearest neighbors in the Discussion section; however, actually, it is the most important work that this paper needs to do to showcase the excellence of their method.
The proposed idea for grouping state by abstraction refining to reduce the branching factor is somewhat similar to [11], which is applied to many continuous action space problems. What’s the main difference between the presented method and their work? A continuous action space can also be viewed as a large action space problem.
This paper tries to justify the computation cost for the nearest neighbor in the last paragraph of Section 3.2, where the cost is derived to be O(log p) or O(log num_children[s, a]) in the parallel time complexity based on the assumption of unlimited (or a large number of) processors. Unfortunately, this assumption is strange and not realistic. It is more important and interesting to justify it through experiments.


Other comments:
The abstraction refining hyperparameters (epsilon_n) are quite different in each experiment. Why? Any guideline to set the hyperparameters? Or, it is a need to do more experiments for different epsilon_n in abstraction refining?
It is also required to analyze the branch factor for abstraction refining and progressive widening for each experiment.

About presentation:
* In line 176, c in R_{s’’}^{c} is undefined.
* What is the epsilon_n used in five by five Go? BTW, it is suggested using “5x5 Go”.
* Some typos:
** Line 40: and and -> and
** Line 98: determinstic -> deterministic
** Line 147: pseuodocode -> pseudocode


**Time Spent Reviewing:**

6 hours

---

> ### Author Response · Authors · 2021-08-10
> **Thanks for the review!**
>
> Thanks for your thorough and insightful comments! We respond to your points below.
>
> - experimental analyses are most based on value errors
>
> We present two experiments based on value errors and two experiments based on expected return. We made this decision because abstraction refining and progressive widening are evaluation algorithms. Thus, even though we are using them downstream for control, the a good way to isolate the difference in performance is actually to examine their policy evaluation performance.
>
> - the nearest neighbors are calculated all based on the geometry of state space (or Euclidean distance). It is very hard to convince people to believe that this would work for some games, like Go.
>
> In the paper we performed experiments on 5x5 Go using the learned network representation, not the naive state space.
>
> - This paper mentions we could use better distance functions for the nearest neighbors in the Discussion section; however, actually, it is the most important work that this paper needs to do to showcase the excellence of their method.
>
> While we understand this perspective, we respectfully disagree that this is the most important work. We feel the most important contribution is showing that, given a good representation the algorithmic contribution is able to leverage this representation to yield stronger performance. That said, we absolutely agree that the representation learning aspect is also worthwhile to pursue and we do offer some evidence that we can learn good representations using a hidden of a PV network in the 5x5 Go experiment. Reference [18] also shows evidence that using a hidden layer of a PV network provides a good metric for behavior similarity for MCTS. Does the reviewer have specific additional experiments in mind?
>
> - The proposed idea for grouping state by abstraction refining to reduce the branching factor is somewhat similar to [11], which is applied to many continuous action space problems. What’s the main difference between the presented method and their work? A continuous action space can also be viewed as a large action space problem
>
> Indeed it is somewhat similar to [11]. However, as you point out, [11] is applied to action spaces whereas our work is applied to state spaces. This leads to significant differences in the methods about when to refine abstractions and how to refine abstractions. [11] adds actions which appear to outperform actions that have already been added to the tree using an epsilon-greedy like sampling scheme over Voronoi cells. We add states (sampled from the transition function) that are sufficiently different from state abstractions that have been used a sufficiently large number of times. Essentially, it is a difference of goals: [11] faces something resembling a continuous bandit problem, whereas we face something like a policy evaluation problem.
>
> - This paper tries to justify the computation cost for the nearest neighbor in the last paragraph of Section 3.2, where the cost is derived to be O(log p) or O(log num_children[s, a]) in the parallel time complexity based on the assumption of unlimited (or a large number of) processors. Unfortunately, this assumption is strange and not realistic. It is more important and interesting to justify it through experiments.
>
> We provide that runtime for **both** non-parallelized and maximally parallelized computation. We think this is a reasonable choice as the true time will lie in between the two.
>
> - The abstraction refining hyperparameters (epsilon_n) are quite different in each experiment. Why? Any guideline to set the hyperparameters? Or, it is a need to do more experiments for different epsilon_n in abstraction refining?
>
> For each experiment we picked a couple of factors that seemed reasonable and used the better forming one. It’s entirely possible that the hyper parameters we picked are suboptimal. We are happy to include a more detailed hyper parameter study for abstraction refining.
>
> - What is the epsilon_n used in five by five Go?
>
> epsilon_n = 200 n^{-0.1} (we will add this to the submission)
>
> We will make changes about presentation as suggested.

---

> > ### Author Response · Authors · 2021-08-17
> > **Following Up**
> >
> > Thanks again for your thorough comments! We wanted to follow up with you to see how our response was received. In particular:
> >
> > - Does the reviewer agree that two value error experiments are worthwhile in light of the fact that the proposed algorithm is a policy evaluation algorithm?
> >
> > - Is it clarified that the 5x5 Go experiments do use a learned embedding for the distance metric?
> >
> > - Is the distinction in purpose and practice between state-wise abstractions and action-wise abstractions clarified?
> >
> > If the reviewer agrees with the above points, would the reviewer be willing to consider raising their score? If not, we would be appreciative of the opportunity to engage in further dialogue with the reviewer.
> >
> > We also want to emphasize that we agree with the reviewer that the paper would benefit from more clarity about the hyper-parameters for abstraction refining. We will include a full hyper-parameter study of abstraction refining in the camera ready copy.
> >
> > EDIT: Also, see discussion with Reviewer MmsP

---

> > > ### Comment · Reviewer_b8VZ · 2021-08-23
> > > **Follow Up**
> > >
> > > After reading the rebuttal, my score still remains the same.
> > >
> > > - It is still unclear how to use a learned embedding in this paper and why this could work? This paper only mentioned, "*The second moving player (white), performs search using the latent state representation of the AlphaZero network as its state representation of the system.*" Specifically, which layer of the AlphaZero network is used for state representation? For example, in the AlphaZero network, let $h_0$ represent the output of policy and value layer, $h_1$ the input for the policy and value layer, $h_2$ the input for the layer which outputs $h_1$, and so on. I don't see which state representation $h_i$ is used (or mixed) in this paper, and also why that would work?
> > >
> > > - I still think it is important to provide further investigations to convince the readers of this work. For example, why is Euclidean distance a proper way to classify (or group) these states? How to build up a good representation that can be easily classified by the Euclidean distance? If a state s1 has a rather small Euclidean distance to state s2 than other states, what is the meaning between these two states? Are the win rates similar, policy similar, or what else could be? In the AlphaZero training, there is no guarantee that states with smaller Euclidean distances can be grouped together. Based on the current results, it seems like the experiments are some kind of trial and error. I think it will strengthen the contribution of this work if more evidence is provided for the readers to build a good representation.
> > >
> > > - As for “runtime analysis”, your answer does not address my question. My point is that for any concerns about runtime you should justify it through experiments (I don't see in this paper), not just using parallel time complexity.

---

> > > > ### Author Response · Authors · 2021-08-23
> > > > **Thanks for following up!**
> > > >
> > > >
> > > > Thanks for your engagement with the review process! We provide some further clarification below:
> > > >
> > > > - We think it may be worth emphasizing that using an internal layer of a network as a state representation is not a contribution of this work. It is widespread practice in deep learning literature to use internal layers of networks as representations and to use distance between internal layer representations. In fact, even in MCTS literature, this has been done before. For example, Memory-Augmented Monte Carlo Tree Search (Xiao 2018), which won AAAI best paper, uses the distance between internal layers of the network to do kernel regression and demonstrates good performance in Go.
> > > >
> > > > - Regarding runtime, we will change the language in the runtime analysis section to incorporate the reviewer's concerns.
> > > >
> > > > - For the AlphaZero experiment, we used the last layer before before the value and policy head split as the representation (h1 in your notation).

---

> > > > > ### Author Response · Authors · 2021-08-27
> > > > > **Hyperparameter study for abstraction refining**
> > > > >
> > > > > As discussed above, we ran a hyper-parameter study for abstraction refining. In the plots, we use epsilon_n = k n^(-alpha). Each column corresponds to a particular k value and the hue corresponds to the alpha value. The x-axis is the number of search iterations. The y-axis is value error for blackjack and expected return for trap.
> > > > >
> > > > > For blackjack we find that smaller alphas perform best with smaller ks and larger alphas perform best with larger ks. Intuitively, this is essentially saying that if the initial notion of similarity is strict, it should only become stricter slowly; but that if the initial notion of similarity is weak, it should become stricter quickly.
> > > > >
> > > > > https://ibb.co/SP6W9N8
> > > > >
> > > > > For trap we find that performance is robust to k as long as a sufficiently small alpha parameter is used. Intuitively, this is saying that AR is relatively robust to the initial notion of strictness as long as it only becomes stricter very slowly.
> > > > >
> > > > > https://ibb.co/JKG4fd2

---

### Official Review · Reviewer_pc96 · 2021-07-17

**Rating:** 6
**Confidence:** 3

**Summary:**

The authors present MCTS with abstraction refining based on the nearest neighbor algorithm.


**Limitations And Societal Impact:**

Yes.


**Main Review:**

MCTS has been a topic of interest in the AI/Machine Learning community over a decade and the authors address an important issue about MCTS. The paper is understandable with some interesting results. However, I have the following comments some of which could be clarified in the rebuttal phase:

1. If I understand the algorithm correctly, how states are abstracted depends on the order of states selected during iterations of MCTS. I wonder how robust abstract refining is due to different orderings of the states.

2. Some researchers have worked on MCTS with state abstractions. Particularly, the following paper might be related to the task they address. I wonder why iterative abstraction refinement was not compared against it.

Jesse Hostetler, Alan Fern, Thomas Dietterich. "Sample-Based Tree Search with Fixed and Adaptive State Abstractions", JAIR 2017.

https://jair.org/index.php/jair/article/view/11096

3. AlphaZero does not account for the size of the territory once a position is likely to be a win/loss -- it only cares about whether to win or loss. Therefore, it is not a good idea to evaluate the algorithm performance based on the territory size. Also, in my understanding, AlphaZero is not combined with search in section 4.4. If this is correct, what would the performance comparison be like if AlphaZero is combined with search?

Miscellaneous

The authors should check the captions in the references (e.g,  Monte Carlo Tree Search which is currently with lowercase).

UPDATE
------------
Thanks for your response. I had discussions with other reviewers and am happy to increase my score.



**Time Spent Reviewing:**

4

---

> ### Author Response · Authors · 2021-08-10
> **Thanks for the review!**
>
> Thanks for your thorough and insightful comments! We respond to your points below.
>
> 1. That is exactly correct. We agree that this would be interesting to see. Though ultimately, what we care about is whether abstraction refining performs well in expectation over orderings, which is what the results in the paper show.
>
> 2. Thanks for this pointer! This work is indeed related and worth discussing. An apparent difference between this work and the algorithms considered in the submission is that this work appears to assume that the set of possible next states is provided to the agent. In contrast, progressive widening and our algorithm only sampling access to the simulator.
>
> 3.
> Size of territory — Our adapted implementation of AlphaZero is trained to maximize the margin of victory (as specified on line 327), rather than maximizing winning percentage, as the original implementation does. We will update the submission to try to make this distinction more explicit.
> Use of search — This is half correct half incorrect. White always uses search whereas Black never uses search and instead just plays according to its policy network. This choice was made so that White faces a stochastic MDP (where the transitions are determined by Black’s moves). It would also in principle be possible to perform the same experiment but use Black’s search policy to determine the transitions. However, this would require determining Black’s search policy at each relevant board state, which would be quite expensive. Hypothetically, if we were to do this, the expected return of both AR and PW would both be much closer to zero. We see no reason that abstraction refining would not continue to outperform progressive widening in this hypothetical.
>
> Captions
>
> Will do! Thanks for the catch.

---

> > ### Author Response · Authors · 2021-08-17
> > **Following Up**
> >
> > Thanks again for your review! We wanted to follow up with you to see where things stand. We believe the response given above addresses all three of the reviewer's points. If so, we would be grateful if the reviewer would consider the possibility of raising their score. If not, would the reviewer be willing to engage in further dialogue regarding any disagreements?
> >
> > We are also happy to include extra experiments re point 1) in the camera ready version of the paper.

---

> > > ### Comment · Reviewer_pc96 · 2021-08-27
> > > **Response**
> > >
> > > Thank you for your response.
> > >
> > > I am happy with your response to my initial review and have a better understanding to your contribution. However, I would still like to see how the discussions with the other reviewers will go.
> > >
> > > For 1), it is better to have numbers in principle, but the paper should be evaluated based with the current experimental results, because the reviewers cannot check the camera ready version once the decision is made.

---

> > > > ### Author Response · Authors · 2021-08-27
> > > > **Thanks for your engagement!**
> > > >
> > > > Re 1), we have finished running the experiments suggested by the reviewer. They are detailed below and available for the reviewer to assess.
> > > >
> > > > For blackjack, we tested whether there was a correlation between the first dealer card sampled and the accuracy of the policy evaluation. The column corresponds to the number of search iterations. The y-axis corresponds to the value error. The x-axis corresponds to the value of the dealer hand.
> > > >
> > > > We did not see a correlation for either of AR and PW.
> > > >
> > > > Results for PW:
> > > > https://ibb.co/gmxHJ5j
> > > >
> > > > Results for AR:
> > > > https://ibb.co/wgryvFp
> > > >
> > > > For the trap problem, we tested whether there was a correlation between whether the first big jump sampled resulted in the agent falling into the trap and the expected return. The x-axis denotes the number of search iterations. The y-axis denotes the expected return. The blue line is the case in which the agent fell on the first big jump sampled and the orange line is the case in which the agent did not fall on the first big jump sampled.
> > > >
> > > > For AR we observed no correlation between whether the first big jump landed in the trap and the expected return.
> > > >
> > > > Results for AR:
> > > > https://ibb.co/NZ0RHy6
> > > >
> > > > For PW we observed that the agent performed significantly worse if the first big jump sampled did not land in the trap. This makes sense because it would lead the agent to believe that taking big jumps is safe and because it reuses the first sample for multiple iterations before taking an additional sample. In contrast, AR samples at every search iteration and the first big jump in which the agent falls into the trap is guaranteed to be added to the tree because the distance between the plateau and the trap is sufficiently large.
> > > >
> > > > Results for PW:
> > > > https://ibb.co/kSLm3Wt

---

### Official Review · Reviewer_MmsP · 2021-07-20

**Rating:** 6
**Confidence:** 4

**Summary:**

The paper proposes a novel extension of MCTS to tackle stochastic state transitions with infinite support. It does so by leveraging the geometry of the state space to "abstract" "similar" states together during the tree building phase. Abstraction refining is introduced to assure that in the limit of infinite samples, the search-tree converges to the full tree, by introducing a schedule on the maximum abstraction distance, based on the number of samples observed.
The authors provide a proof of asymptotic convergence of abstraction refining in the policy evaluation setting.
A brief discussion is done on the choice of distance function to employ during the state abstraction.
Finally, an experimental campaign is conducted in several evaluation and control tasks, that show that abstraction refining consistently outperforms it's main competitor (progressive widening) in handling stochastic transitions with infinite or very large support.

**Limitations And Societal Impact:**

Main limitation is:
The need for a carefully thought distance measure.

**Main Review:**

The authors tackle the really important topic of planning in environments with stochastic transitions and infinite next-state support. The introduce abstraction refining, as a way to progressive build the full (infinite) search tree, by leveraging the geometry of the state space.
The authors, mainly compare their work, with progressive widening, the only method (apart from open-loop planning) able to tackle these kind of problem.
They provide a proof of convergence in the policy evaluation case, but do not provide a proof of convergence in the control case, which is a disadvantage compared with progressive  widening, which when used together with UCT, for an appropriate choice of schedule for the \alpha parameter is shown to converge [1].

The authors provide a brief consideration of the additional computational costs of the method which show a limited additional cost, based on the dimensionality of the state space (or learned feature space) used to compute the distance.

The authors provide some experimental results both in the policy evaluation and in the control settings. The proposed method consistently shows better results than progressive widening in all the baselines. Nevertheless, the choice of baselines seems rather odd. While some simple baselines are chosen to study in detail the performance of abstraction refining (like the toy MDP, and the Trap environment), the authors also evaluate in a more complex scenario of 5x5 GO, which intrinsically has no stochasticity, bu the stochasticity is induced by considering the environment as a single player, and fixing the stochastic policy of the opponent. While this makes the environment stochastic, it still does not capture the main contribution of the paper, the capacity to handle larger (or infinite) branch factors of the tree, since the authors report that the best performing algorithm in this setting was Vanilla MCTS (in the appendix) and they were force to introduce an additional source of stochasticity to require the use of PW or AR.
The authors have a brief discussion of the choice of distance measure, since it is a crucial component of AR, but a more thorough experimental campaign is needed. Three of the presented domains, were simple enough to use Euclidean distance. In the 5x5 Go, the distance was employed on a learned state embedding, and nevertheless failed to improve on vanilla MCTS without an additional noise.

I think the paper tackles an extremely important problem but more careful consideration of the distance measures needs to be adressed, at least experimentally.
[1] Adrien Couetoux. Monte Carlo Tree Search for Continuous and Stochastic Sequential Decision Making
Problems. Data Structures and Algorithms [cs.DS]. Université Paris Sud - Paris XI, 2013.

======== Update ==≠====
 I have had a thorough discussion with the authors, during which the authors have agreed to the limitations that I pointed out, and also have conducted some experiments that demonstrate these limitations. In my opinion, it is ok to have limitations but they should be made clear, and especially better if demonstrated empirically. As a result, I will raise my score from 5 to 6, but this if the authors include a final experiment promised in the last rebuttle. While I am aware that as reviewers we cannot enforce the addition of the experiment, the authors have already made 3 new experiments during the rebuttle so I would be inclined to 'trust' the authors for this final experiment.

**Time Spent Reviewing:**

4.5

---

> ### Author Response · Authors · 2021-08-10
> **Thanks for the review!**
>
> Thanks for your thorough and insightful comments! We respond to your points below
>
> - Proof on convergence (disadvantage compared to progressive widening)
>
> There may be a misunderstanding regarding the proof of convergence. Abstraction refining (like state-wise progressive widening) is an algorithm for policy evaluation, not control. Given the result in the submission (that abstraction refining converges in the policy evaluation case) it follows that abstraction refining converges when combined with control algorithms with convergence guarantees.
>
> - Choice of baselines (benchmarks(?)) “Simple enough to use Euclidean distance” “more careful consideration of distance measure is needed”
>
> As the reviewer points out, our choice of benchmarks allowed us to carefully analyze the performance of abstraction refining compared to that of progressive widening across many hyper parameter settings, despite that each hyper parameter setting required hundreds or thousands of runs (far more than required for analogous deterministic experiments).
>
> The submission attempts to show that, given a good state representation, abstraction refining can outperform progressive widening. For that reason, we specifically sought out benchmarks for which Euclidean distance over a naive state representation was reasonable in order to isolate the algorithmic contribution from the representation learning problem. However, that Euclidean distance is reasonable does not necessarily speak to the environment’s simplicity.
>
> Still, we wholeheartedly agree that work on representation learning is an important direction. The 5x5 Go experiments suggest that a learned network representation can be effective. Reference [18] also shows evidence that the learned network representation is useful metric for comparing states when performing MCTS.
>
> Are there other specific benchmarks the reviewer has in mind?
>
> - the authors also evaluate in a more complex scenario of 5x5 GO, which intrinsically has no stochasticity, bu the stochasticity is induced by considering the environment as a single player, and fixing the stochastic policy of the opponent. While this makes the environment stochastic, it still does not capture the main contribution of the paper, the capacity to handle larger (or infinite) branch factors of the tree, since the authors report that the best performing algorithm in this setting was Vanilla MCTS (in the appendix) and they were force to introduce an additional source of stochasticity to require the use of PW or AR
>
> While it is true that additional stochasticity was required to motivate the use of PW and AR, we do not think that that fact makes the experimental results unworthwhile. In particular, it is clear from this experiment that PW can be brittle to the problem setting in a sense that AR is not.

---

> > ### Author Response · Authors · 2021-08-17
> > **Following Up**
> >
> > We wanted to follow up to get feedback on the reception of the comments above. In particular:
> >
> > - Is the reviewer in agreement on the clarification above regarding the proof of convergence?
> >
> > - Does the reviewer find the reasoning given above regarding the choice of benchmarks compelling?
> >
> > - Does the reviewer agree that 5x5 Go experiments offer some value, as argued above?
> >
> > If yes to the above, would the reviewer be willing to consider raising their score?
> >
> > If no, would the reviewer be willing to engage in further discussion about the relevant disagreements?
> >
> > We would also be grateful for specific suggestions about distance measure experiments that the reviewer would like to see.
> >
> > Thanks again for your thoughtful comments!

---

> > > ### Comment · Reviewer_MmsP · 2021-08-17
> > > **Follow Up**
> > >
> > > Is the reviewer in agreement on the clarification above regarding the proof of convergence?
> > > Not completely. While I agree that the proposed algorithm is a policy evaluation algo., as the authors also mention in the paper, MCTS is nearly never used for this purpose. PW is also a policy evaluation algo, but the authors of PW provide a proof of convergence when used together with UCT. Specifically, they provide a schedule of the alpha parameter as a function of the tree depth. So the theoretical value of proving convergence with of AR with UCT would be a theoretical schedule for the \epsilon parameter and a comparison of AR and PW with the theoretical values for \epsilon and \alpha. So I do not agree with the authors when they say that this type of analysis is not very interesting.
> > >
> > > Does the reviewer find the reasoning given above regarding the choice of benchmarks compelling
> > > I agree this the above reasoning, that the benchmarks are interesting. What I proposed though was an additional experiment, where the Euclidean distance is not enough to choose whether to abstract or not. This would be interesting as it would show if AR could be detrimental for the performance when the distance measure is not good (As I suspect because it would most likely affect the exploration of the tree, focusing in. suboptimal regions of the tree). This could be done in a simple Cliffworld environment where states close to the border of the cliff need to be differentiated from states at the border.
> > >
> > > Does the reviewer agree that 5x5 Go experiments offer some value, as argued above?
> > >
> > > While I agree that the experiment is somehow significant, what I meant is that the main contribution of the work, is a method capable of handling environments with stochastic transitions and continuous state spaces, so I would also expect an experiment in this setting (maybe a navigation task with stochastic transitions or a mujoco benchmark with noise). 5x5 Go is a discrete environment, even though with many states, they are still discrete and moreover the cardinality of the next state distributions is even more limited (that is why also plain alphazero works well). A motivating example should show a scenario where AZ plain fails and AZ with AR succeeds in solving the domain.
> > >
> > > With the current answers I still think that the score I gave is appropriate.

---

> > > > ### Author Response · Authors · 2021-08-18
> > > > **Re Follow Up**
> > > >
> > > > Thanks very much for your prompt and actionable response.
> > > >
> > > > > Convergence Proof
> > > >
> > > > We are slightly unsure of what the reviewer means here. Regarding the schedule of epsilon, any sequence satisfying the properties on lines 168-170 will guarantee convergence under the assumptions made in Proposition 1. If the reviewer is talking about the lack of a rate of convergence, then we agree that this is a disadvantage compared to the result in [1]. It would be possible to provide a rate of convergence for AR, but it would not be very informative of behavior in practice without additional assumptions about the quality of the distance function.
> > > >
> > > > > Euclidean Distance
> > > >
> > > > Ah, thanks for clarifying, we misunderstood before. We agree it is important to emphasize that Euclidean distance is not usually a good choice and also that abstraction refining is not a good choice if there is not a good distance function available. To illustrate this point, we ran abstraction refining over a sweep of hyperparameters on the Trap problem, using a distance metric that only provides the agent information about its x-position, rather than its (x,y) position, as was done in the submission. Effectively, this means the agent is not able to see whether or not it has fallen into the trap. As illustrated in the plot linked below, abstraction refining performs quite poorly with this distance function with the hyperparameters used in the submission. A more careful choice of hyperparemeters avoids catastrophic failure, but is unable to match the performance of abstraction with the distance function used in the submission.
> > > >
> > > > https://ibb.co/kc60jY2
> > > >
> > > > In the plot, we use epsilon_n = k n^(-alpha). The column is the value of k and the hue corresponds to the value of alpha. The values on the y-axis are directly comparable to the Trap results shown in the submission.
> > > >
> > > > > Stochastic transition, continuous state spaces, and learned representation
> > > >
> > > > We are currently working on putting together an experiment with these properties, but will require more time than for the additional Trap experiments above. We will post again later when we have results.
> > > >
> > > > Thanks again for your engagement!

---

> > > > > ### Author Response · Authors · 2021-08-23
> > > > > **Additional Experiment**
> > > > >
> > > > > We have attached an additional experiment below. For the experiment, we used the openAI gym pendulum environment, with discretized actions and stochasticity in the form of actuator noise. The actuator noise comes in two forms. First, the agent has a “fat finger” in the sense that, with probability epsilon, a random action is selected instead of the intended action. Second, uniform mean zero noise is added to the continuous representation of the selected action. We performed the experiment in two steps. In the first step, we ran DQN with a four layer network on the environment. The average reward of DQN is (-5.642 +/- 0.04). In the second step, we ran AZ, using a Boltzmann distribution over DQN Q-values as the policy prior, the maximum DQN Q-value as the state value, and the learned network representation for the distance function of AR.
> > > > >
> > > > > The results are shown in the plot below. The x-axis shows the number of search iterations (values are measured at 50, 100, and 200). The y-axis shows the average reward. Evaluating algorithm performance was quite costly in the sense that we required a very large number of games to achieve a high level of certainty. (The lines below are each averages over 100000 games.) As a result, it was not possible to do a full hyper-parameter for each algorithm. Instead, we ran 1000 games for each of AR and PW over sets of 5 hyper-parameters and selected the best performing for each to run a full 100000 games on. With the selected hyper-parameters, we find that AR outperforms PW with statistical significance, and that PW outperforms AZ with statistical significance.
> > > > >
> > > > > https://ibb.co/5LL4RGW
> > > > >
> > > > > Note that this experiment satisfies the criteria above (continuous state space, realistic stochasticity, and a learned representation for abstraction refining). It also satisfies the reviewer’s suggestion of an example in which AR outperforms plain AZ.

---

> > > > > > ### Comment · Reviewer_MmsP · 2021-08-24
> > > > > > **Follow Up Experiments**
> > > > > >
> > > > > > I thank the authors for the additional experiments.
> > > > > > 1.
> > > > > > I am still concerned with the theoretical part, as the authors continue to claim that the algorithm would converge for any schedule of \epsilon when used in a control setting. I do not agree. The authors have proved convergence in the policy evaluation setting, but to prove convergence in the control setting, for example with a UCT like selection policy, the authors would need to show that the AR generates a non-stationarity of the return process in the internal nodes of the tree still satisfies assumption 1 in [1], wihch would make the UCB selection converge to the optimal choice in each node of the tree.
> > > > > > I do not agree that proving convergence in the policy evaluation setting makes the results extremely straightforward to extend to the control setting as stated in the paper. If they are straightforward and I fail to see it, I would invite the authors to include the result in the paper (or at least an outline of the proof).
> > > > > >
> > > > > > 2. While I thank the authors for the additional experiments, I find odd the choice of pretraining the AZ policy by using DQN. Why not use AZ directly with AR during the search? Is this not an intended application of AR?  Using AR only after pretraining the policy, and using it from the beginning when the representation is random would produce extremely different results. Namely the question here would be, does using AR with the internal network representation during the learning process negatively affect the learning process of AZ?
> > > > > >
> > > > > > I am willing to raise my score if these two final answers are answered.
> > > > > >
> > > > > > [1] Kocsis, L., Szepesvari, C., & Willemson, J. (2006). Improved Monte-Carlo Search.

---

> > > > > > > ### Author Response · Authors · 2021-08-26
> > > > > > > **Follow Up**
> > > > > > >
> > > > > > > > Convergence
> > > > > > >
> > > > > > > Thanks for continuing to push this point. We agree with the reviewer and retract our earlier statements. We are looking into whether this would be possible. We will clarify the text to make it clear that convergence in the policy evaluation case does not directly imply convergence with UCT.
> > > > > > >
> > > > > > > > Why not use AZ at train time?
> > > > > > >
> > > > > > > The AlphaZero implementation we used for the Go experiments is wedded to the OpenSpiel codebase and integrating the AlphaZero training logic into our other codebase would’ve required large changes. When we realized that algorithm evaluation would take a significant amount of time, we decided to take the easier implementation route of only doing search at test time to hasten the turnaround time of the experiment. This paradigm is common in imperfect information games (e.g., DeepStack, Libratus, Pluribus, SPARTA), though admittedly not as common in perfect information ones. We will include experiments with train-time AlphaZero in a camera ready version if the paper is accepted.
> > > > > > >
> > > > > > > > AZ with AR at train time
> > > > > > >
> > > > > > > We think this would be a cool use case! However, as the reviewer points out, there is an apparent drawback of this approach in that the representation at the beginning of training is not a meaningful measurement of state similarity. We strongly suspect that this would indeed negatively affect the learning process early in training. One possible amelioration would be to use a sequence of {epsilon_n} that requires a very strict notion of similarity early on in training and use {epsilon_n} that require progressively less strict notions of similarity as training goes on. We will also include results comparing AZ, AZ with PW at training, and AZ with AR at training if accepted.
> > > > > > >
> > > > > > > > Overall thoughts on review process
> > > > > > >
> > > > > > > We wanted to thank the reviewer for the extremely high quality review process. The reviewer demonstrated a very strong of understanding of the relevant material and made numerous constructive comments. We wish the review process was like this more often.

---

### Decision · Program_Chairs · 2021-09-27

**Decision:**

Accept (Poster)

**Comment:**

This paper generated a good amount of discussion between the authors and reviewers, which helped resolve some issues in the original reviews. Certain issues remain, but the paper does seem to be making a reasonable and novel contribution in light of those issues.

The reviewers have worked hard to suggest improvements and the authors have already indicated the inclusion of new experiments and adjustments to the text. It is expected that the authors will follow through on these promises.

Finally, I would like to point out that the authors have not properly characterized the related paper pointed out by one of the reviewers.

Jesse Hostetler, Alan Fern, Thomas Dietterich. "Sample-Based Tree Search with Fixed and Adaptive State Abstractions", JAIR 2017

The authors indicate in their response that the paper assumes more than a sample-based model, but that is not accurate. It does not assume access to the entire set of next states. Also note that the algorithm in the experiments of that paper is based on forward-search sparse sampling, which is a trajectory-sampling algorithm in the spirit of MCTS/UCT. Thus a modification to MCTS is not as distance as the authors may think.